# Adaptive Spatial-temporal Inception Graph Convolution Networks for Multi-step spatial-temporal Data Forecasting

## Abstract

Spatial-temporal data forecasting is of great importance for industries such as telecom network operation and transportation management. However, spatial-temporal data are inherent with complex spatial-temporal correlations and behaves heterogeneities among the spatial and temporal aspects, which makes the forecasting remain as a very challenging task though recently great work has been done. In this paper, we propose a novel model, Adaptive Spatial-Temporal Inception Graph Convolution Networks (ASTI-GCN), to solve the multi-step spatial-temporal data forecasting problem. The model proposes multi-scale spatial-temporal joint graph convolution block to directly model the spatial-temporal joint correlations without introducing elaborately constructed mechanisms. Moreover inception mechanism combined with the graph node-level attention is introduced to make the model capture the heterogeneous nature of the graph adaptively. Our experiments on three real-world datasets from two different fields consistently show ASTI-GCN outperforms the state-of-the-art performance. In addition, ASTI-GCN is proved to generalize well.

## 1 Introduction

Spatial-temporal data forecasting has attracted attention from researchers due to its wide range of applications and the same specific characteristics of spatial-temporal data. Typical applications include mobile traffic forecast (He et al., 2019), traffic road condition forecast (Song et al., 2020; Yu et al., 2017; Guo et al., 2019; Zheng et al., 2020; Li et al., 2017), on-demand vehicle sharing services passenger demand forecast (Bai et al., 2019) and geo-sensory time series prediction (Liang et al., 2018) etc. The accurate forecast is the foundation of many real-world applications, such as Intelligent Telecom Network Operation and Intelligent Transportation Systems (ITS). Specifically, accurate traffic forecast can help transportation agencies better control traffic scheduling and reduce traffic congestion; The traffic volumes prediction of the wireless telecommunication network plays an important role for the network operation and optimization, for example, it can help to infer the accurate sleep periods (low traffic periods) of the base stations to achieve energy saving without sacrificing customer experience.

However, as we all know, accurate spatial-temporal data forecasting faces multiple challenges. First, it is inherent with complex spatial-temporal correlations. In the spatial-temporal graph, different neighbors may have different impacts on the central location at the same time step, as the bold lines shown in Figure1, which called spatial correlations.Different historical observations of the same location influence the future moments of itself variously due to temporal correlations. The observations

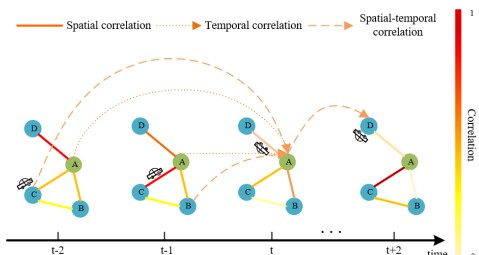

Figure 1: Spatial-temporal correlations.

of different neighbors at historical moments can directly affect the central node at future time steps due to the spatial-temporal joint correlations. As shown in Figure1, the information of the spatial-temporal network can propagate along the spatial and temporal dimensions simultaneously, and the

transmission process can be discontinuous due to complex external factors, which result in spatial-temporal joint correlations of the spatial-temporal data in a short period.

Spatial-temporal data is heterogenous in both spatial and temporal dimensions (Song et al., 2020). Nodes in different regions of the graph have various properties and local spatial structures, so the corresponding data distribution can be different. For example, the traffic flow distribution of urban and suburban areas are quite different, while the traffic of urban area is denser and that of suburban area is relatively sparse. Besides, the traffic flow in the same region also exhibit heterogeneity in different time periods. For example, the mobile traffic in business district would decrease at night compared to the daytime, while it's opposite in the residential district. In addition, multi-step time series forecasting is often accompanied by error accumulation problem. Typical methods like RNNs often cause error accumulation due to iterative forecasting, leading to rapid deterioration of the long-term prediction accuracy. (Yu et al., 2017; Zheng et al., 2020).

Most of the previous work is mainly to solve the above challenges. To model the spatial-temporal dependency, STGCN (Yu et al., 2017) and DCRNN (Li et al., 2017) extract spatial and temporal correlations separately. ASTGCN (Guo et al., 2019) introduced spatial and temporal attention to model the dynamic spatial and temporal correlations. STG2Seq (Bai et al., 2019) aimed at using GCN to capture spatial and temporal correlations simultaneously. But they all didn't consider the spatial-temporal joint correlations and heterogeneity. Different from the above methods, STSGCN (Song et al., 2020) used multiple local spatial-temporal graphs to model the spatial-temporal synchronous correlations and spatial-temporal heterogeneity of the local adjacent time steps. But STSGCN can only model the spatial-temporal synchronous correlations of its defined local spatial-temporal graphs and it is equipped with complex structure.

In this paper, we propose a novel model called ASTI-GCN, Adaptive spatial-temporal Inception Graph Convolutional Networks, to address the above issues with multi-step spatial-temporal data forecasting. We propose the spatial-temporal joint convolution to directly model the spatial-temporal joint correlations without introducing elaborately constructed mechanisms. And we introduce the inception mechanism to build multi-scale spatial-temporal features to adapt to graph nodes with different properties. Then, to achieve the heterogeneity modeling, we construct the spatial-temporal Inception Graph Convolution Module, which combined the spatial-temporal inception mechanism with the graph attention to build the adaptive ability of graph nodes with different properties. After multiple spatial-temporal inception-GCMs, two decoder modules named sequence decoder and short-term decoder are designed to directly establish the relationships between the historical and future time steps to alleviate error accumulation.

Overall, our main contributions are summarized as follows:

- We propose a novel spatial-temporal joint graph convolution network to directly capture spatial-temporal correlations. Moreover, we introduce inception with graph attention to adaptively model the graph heterogeneity.

- We propose to combine the sequence decoder and short-term decoder together for multi-step forecasting to model direct relationships between historical and future time steps to alleviate the error propagation.

- We evaluate our model on three real-world datasets from two fields, and the experimental results show that our model achieves the best performances among all the eight baselines with good generalization ability.

## 2 RELATED WORK

Spatial-temporal data information can be extracted using the deep learning method from European space, such as ConvLSTM (Xingjian et al., 2015), PredRNN (Gowrishankar & Satyanarayana, 2009) and so on. However, most of the spatial-temporal data in real scenes are graph data with complex and changeable relationships.Common timing prediction models, such as HA and ARIMA (Williams & Hoel, 2003), cannot be simply migrated to such scenarios. Graph based methods like DCRNN (Li et al., 2017) modeled traffic flow as a diffusion process on a directed graph. Spatial dependencies and temporal dependencies are captured by bidirectional random walk and DCGRU based

encoder-decoder sequence to sequence learning framework respectively. STGCN (Yu et al., 2017) constructed an undirected graph of traffic network, which is combined with GCN and CNN to model spatial and temporal correlation respectively. ASTGCN (Guo et al., 2019) innovatively introduced attention mechanisms to capture dynamic spatial and temporal dependencies. Similarly, GMAN (Zheng et al., 2020) used temporal and spatial attention to extract dynamic spatial-temporal correlations with spatial-temporal coding. The above models extract spatial-temporal correlation with two separate modules, which cannot learn the influence of neighbor node at the same time and the influence of center node at the historical moment simultaneously. To address this problem, Bai et al. (2019) proposed STG2seq to learn the influence of spatial and temporal dimensions at the same time, which is a purely relies on graph convolution structure. However, all the above methods fail to take the heterogeneity of spatial-temporal data into account, that is, the scope of each node influencing its neighbor nodes at future time steps is different. To solve this problem, Song et al. (2020) proposed STSGCN with multiple modules for different time periods to effectively extract the heterogeneity in local spatial-temporal maps. However, this method pays more attention to local information and lacks of global information extraction. Besides, STSGCN runs slowly due to too many parameters.

Therefore, we propose an Adaptive spatial-temporal Inception Graph Convolutional Networks The Temporal and spatial correlations are extracted simultaneously by spatial-temporal convolution, and the node heterogeneity is modeled by Inception mechanism. At the same time, considering the different influences of each node and time step, the attention mechanism is introduced to adjust the influence weight adaptively.

## 3 METHODOLOGY

### 3.1 PRELIMINARIES

In this paper, we define $\mathcal{G} = (V, E, A)$ as a static undirected spatial graph network. $V$ represents the set of vertices, $|V| = N(N$ indicates the number of vertices). $E$ is the set of edges representing the connectivity between vertices. $A \in \mathbb{R}^{N \times N}$ is the adjacency matrix of the network graph $\mathcal{G}$ where $A_{v_i, v_j}$ represents the connection between nodes $v_i$ and $v_j$. The graph signal matrix is expressed as $X_t \in \mathbb{R}^{N \times C}$, where $t$ denotes the timestep and $C$ indicates the number of features of vertices. The graph signal matrix represents the observations of graph network $\mathcal{G}$ at time step $t$.

**Problem Studied** Given the graph signal matrix of historical $T$ time steps $\chi = (X_{t_1}, X_{t_2}, \ldots, X_{t_T}) \in R^{T \times N \times C}$, our goal is to predict the graph signal matrix of the next $M$ time steps $\hat{Y} = \left( \hat{X}_{t_{T+1}}, \hat{X}_{t_{T+2}}, \ldots, \hat{X}_{t_{T+M}} \right) \in R^{M \times N \times C}$. In other words, we need to learn a mapping function $F$ to map the graph signal matrix of historical time steps to the future time steps:

$$\left( \hat{X}_{t_{T+1}}, \hat{X}_{t_{T+2}}, \cdots, \hat{X}_{t_{T+M}} \right) = \mathcal{F}_\theta \left( X_{t_1}, X_{t_2}, \cdots, X_{t_T} \right) \tag{1}$$

where $\theta$ represents learnable parameters of our model.

### 3.2 ARCHITECTURE

The architecture of the ASTI-GCN proposed in this paper is shown in Figure 2(a). The main ideas of ASTI-GCN can be summarized as follows: (1) We propose spatial-temporal joint graph convolution to directly extract the spatial-temporal correlations; (2) We build the Spatio-Temporal Inception Graph Convolutional Module (STI-GCM) to adaptively model the graph heterogeneity; (3) We use short-term decoder combined with sequence decoder to achieve accurate multi-step forecast.

### 3.3 SPATIAL-TEMPORAL INCEPTION-GCM

**Spatial-temporal joint graph convolution**

In order to extract spatial-temporal correlations simultaneously, we propose spatial-temporal joint graph convolution. In this paper, we construct spatial-temporal joint graph convolution based on graph convolution in the spectral domain. The spectral graph convolution implemented by using the graph Fourier transform basis which is from eigenvalue decomposition of the Laplacian matrix ($L$)

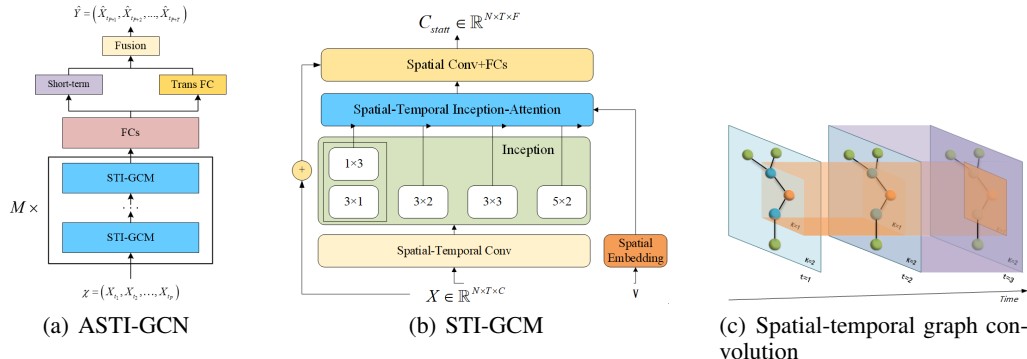

(a) ASTI-GCN

(b) STI-GCM

(c) Spatial-temporal graph convolution

Figure 2: (a) The entire architecture of ASTI-GCN. (b)The spatial-temporal inception graph convolutional module (the convolutional kernel size in the inception block can be changed according to the data). (c) A schematic diagram of spatial-temporal joint graph convolution.

to transform the graph signals from spatial into the spectral domain. But the computation cost of the eigenvalue decomposition of $L$ is expensive when the graph is large. To reduce the number of parameters and the computation complexity, Chebyshev polynomial $T_k(x)$ is used for approximation. The spectral graph convolution can be written as (Yu et al., 2017; Guo et al., 2019; Kipf & Welling, 2016):

$$\Theta *_{\mathcal{G}} x = \Theta(L) x \approx \sum_{k=0}^{K-1} \theta_k T_k\left(\tilde{L}\right) x \tag{2}$$

where $*_{\mathcal{G}}$ is graph convolution operator, $\Theta$ is graph convolution kernel, $x \in \mathbb{R}^N$ is the graph signal, $T_k\left(\tilde{L}\right) \in \mathbb{R}^{N \times N}$ is the Chebyshev polynomial of order $k$ with the scaled Laplacian $\tilde{L} = \frac{2}{\lambda_{max}} L - I_N$ ($L$ is the graph Laplacian matrix, $\lambda_{max}$ is the largest eigenvalue of $L$, $I_N$ is identity matrix) (Yu et al., 2017). $\theta_k$ is the coefficient of the $k$-th order polynomial.

Based on spectral domain graph convolution, we propose spatial-temporal joint graph convolution. First, these $K$-hop $T_k\left(\tilde{L}\right)$ are concatenated as the furthest receptive field in the spatial dimension. Then we construct the spatial-temporal joint graph convolution kernel $\Theta_{s,t}$, $\Theta_{s,t} \in \mathbb{R}^{K_t \times K_s \times C \times F}$, where $K_t$ represents the kernel size in the temporal dimension and $K_s$ represents the kernel size in the spatial dimension, $C$ is the input feature dimensions, $F$ is the number of filters. So the kernel $\Theta_{s,t}$ has the local spatial-temporal receptive field of $K_t \times K_s$, and $K_s$ should be lower than $K$( which can be written as $K_s < K$), because of the largest graph convolution perceived field of $K$-hop. And the spatial-temporal joint graph convolution can be formulated as:

$$\mathbf{T_K}\left(\tilde{L}\right) = Concat(T_0\left(\tilde{L}\right), T_1\left(\tilde{L}\right), \ldots, T_{K-1}\left(\tilde{L}\right)) \tag{3}$$

$$X_{out} = \Theta_{s,t} * X = \Theta_{s,t} \mathbf{T_K}\left(\tilde{L}\right) X \tag{4}$$

where $\mathbf{T_K}\left(\tilde{L}\right) \in \mathbb{R}^{K \times N \times N}$ is the concatenation of all Chebyshev polynomials in (K-1) hop. $*$ is the convolution operation between $\Theta_{s,t}$ and $X$, $X \in \mathbb{R}^{N \times T \times C}$ is the spatial-temporal signal of the input graph, T is the input time steps. After the spatial-temporal joint graph convolution, the output can be written as $X_{out} \in \mathbb{R}^{N \times (T-K_t+1) \times (K-K_s+1) \times F}$. Besides, the neighbors have various influences on the central node, so we implement a learnable spatial mask matrix $W_{mask} \in \mathbb{R}^{N \times N}$ (Song et al., 2020) to adjust the graph adjacency relationship for assigning weights to different neighbors.

**Spatial-temporal inception-attention**

Different from the images, each node of the spatial-temporal graph usually represents a road or eNodeB etc. Then, affected by external factors like geographic location and surrounding environment, the spatial-temporal data properties of different nodes are various, namely the heterogeneity. To solve this problem, an intuitive method is to learn different models for each node, but this method could cause extensive parameters and maintain low generalization ability. So we take another way in this paper. We understand that heterogeneity is manifested in the differences of local spatial-temporal receptive fields of each node of the graph which result from nodes' various properties and local spatial structures. Inspired by (Song et al., 2020; Zheng et al., 2020; Zhou et al., 2018; Vaswani et al., 2017), we apply a learnable graph node embedding $S_e \in R^{N \times E}$ to represent the properties of each node. Meanwhile, we introduce inception (Szegedy et al., 2015) to extract multi-scale spatial-temporal correlations through spatial-temporal joint graph convolution. Then, we combine the graph attention with inception to achieve node-level attention for modeling the heterogeneity.

Firstly, we implement inception, as shown in Figure 2(b). For example, the $3 \times 2$ block represents that it involves the kernel $\theta_{s,t} \in R^{3 \times 2 \times C \times F}$, which means it can extract the spatial-temporal correlations of the node itself and its neighbors in the three adjacent time steps by one layer, which is needed by two layers STSGCM in STSGCN (Song et al., 2020). We use the padding method when implement inception, so after $B$ branches, the output of the inception module can be $C_{out} \in R^{N \times T \times K \times (F \times B)}$, where we set the number of output filters of each branch to be the same. Then we combined with the graph node attention, which has being widely used (Vaswani et al., 2017). We use the $Q = S_e W_q$, $W_q \in R^{E \times F}$ to get the queries of graph nodes. For each branch in inception, we apply the idea of SKNET (Li et al., 2019) to do global pooling $C_{gl} = \sum_{i=1}^{T} \sum_{j=1}^{K} C_{out}, C_g \in R^{N \times (F \times B)}$ as the corresponding keys of the branches (one can also use $W_k \in R^{F \times F}$ to do transform as (Vaswani et al., 2017), here we omit it for simplicity). Then we compute the attention by $S = QK^T$ (Vaswani et al., 2017). Take a graph node $v_i$ as an example, $S_{v_i,b} = \frac{q_{v_i} \bullet c_{g,v_i,b}}{\sqrt{F}}$, $q_{v_i} \in R^{1 \times F}$, $c_{g,v_i,b} \in R^{1 \times F}$ denotes the attention of $v_i$ and each branch $C_{g,v_i,b}$ which represents the corresponding spatial-temporal receptive field perception. Then we concatenate the inception branch results adjusted by attention score to obtain the output. The calculation can be formulated as follows:

$$\alpha_{v_i,b} = \frac{\exp(S_{v_i,b})}{\sum_{b_c=1}^{B} \exp(S_{v_i,b_c})} \tag{5}$$

$$\text{Att}_{v_i} = ||_{b=1}^{B} \left\{ \alpha_{v_i,b} \cdot C_{out,v_i,b} \right\} \tag{6}$$

where $\alpha_{v_i,b} \in R$, $C_{out,v_i,b} \in R^{T \times K \times F}$, $\text{Att}_{v_i} \in R^{T \times K \times F \times B}$ represent the output of node $v_i$ from inception-attention block. Therefore, the final output of spatial-temporal inception-attention block is $C_{att} \in R^{N \times T \times K \times F \times B}$.

**STI-GCM output Layer**

Then, we use spatial convolution to generate the output of STI-GCM. We first reshape $C_{att}$ into $C_{att} \in R^{N \times T \times K \times (F \cdot B)}$. Next, the learnable weight matrix $W_s \in R^{K \times (F \cdot B) \times (F \cdot B)}$ is used to convert $C_{att}$ to $C_{satt} \in R^{N \times T \times (F \cdot B)}$. We also implement SE-net (Hu et al., 2020) to model the channel attention. Finally, the output is converted to $C_{statt} \in R^{N \times T \times F}$ using the full connection layer with $W_o \in R^{(F \cdot B) \times F}$. And the process can be formulated as $C_{statt} = C_{att} W_s W_o$.

### 3.4 Fusion Output Module

The iterative forecasting can achieve high accuracy in short-term forecasting, but in long term prediction, its accuracy would decrease rapidly due to the accumulation of errors. So, we propose the sequence transform block named sequence decoder, to build direct relationships between historical and future multi time steps. Since different historical time steps have different effects on different future time steps, we introduce temporal attention by $\hat{Y}_{s\_out} = C_{statt} W_F W_t (W_F \in R^{F \times 1}, W_t \in R^{t \times M})$ to adjust the corresponding weights of historical temporal features. At last, in order to benefit from both, we adopt the fusion of iterative prediction and sequence transform prediction results:

$$\hat{Y} = g(\hat{Y}_{m\_out}, \hat{Y}_{s\_out}) \tag{7}$$

where $\hat{Y}_{m\_out}, \hat{Y}_{s\_out}$ represent the prediction of the short-term decoder and sequence decoder prediction result respectively, $g(\bullet)$ represents the weighted fusion function, $\hat{Y}$ is the final multi-step forecasting result.

In this paper, we use the mean square error (MSE) between the predicted value and the true value as the loss function and minimize it through backpropagation.

$$\mathrm{L}(\theta) = \frac{1}{\tau} \sum_{i=t+1}^{i=t+\tau} (Y_i - \hat{Y}_i)^2 \tag{8}$$

where $\theta$ represents all learnable parameters of our model, $Y_i$ is the ground truth. $\hat{Y}_i$ denotes the model's prediction of all nodes at time step $i$.

## 4 EXPERIMENT

We evaluate ASTI-GCN on two highway datasets (PEMSD4, PEMSD8) and one telecommunications dataset. The traffic datasets come from the Caltrans Performance Measurement System (PeMS) (Chen et al., 2001). The network traffic dataset comes from the mobile traffic volume records of Milan provided by Telecom Italia (G et al., 2015).

**PEMSD4**: PEMSD4 comes from the Caltrans Performance Measurement System (PeMS) (Chen et al., 2001). It refers to the traffic data in San Francisco Bay Area from January to February 2018.

**PEMSD8**: PEMSD8 comes from the Caltrans Performance Measurement System (PeMS) (Chen et al., 2001). It is the traffic data in San Bernardino from July to August 2016.

**Mobile traffic**: It contains records of mobile traffic volume over 10-minute intervals in Milan where is divided into 100*100 regions. We use the data in November 2013.

We divide the road traffic datasets at a ratio of 6:2:2 for train set, validation set and test set. For the mobile traffic dataset, we use the first 20 days of data to train models and the next 10 days for testing. All datasets are normalized by standard normalization method before training and renormalized before evaluation. We implemented our model in Python with TensorFlow 1.14. In the experiment, we use one-hour historical data to predict the data of the next hour. The hyperparameters of ASTI-GCN are determined by the performance on the validation set. The best size of spatial-temporal kernels is set to $3 \times 1$, $1 \times 3$, $5 \times 2$, $3 \times 2$, $2 \times 3$, which are used in 4 spatial-temporal inception layers respectively. We evaluate the performance of our network with three evaluation metrics: Mean Absolute Error (MAE), Root Mean Square Error (RMSE) and Mean Absolute Percentage Error (MAPE). For the road traffic datasets, if two nodes are connected, the corresponding value in the adjacency matrix is set to 1, otherwise 0. For the Milan dataset, Spearman correlation coefficient was used to define the adjacency matrix. We compare ASTI-GCN with other widely used forecasting models, including HA, ARIMA (Williams & Hoel, 2003), LSTM (Hochreiter & Schmidhuber, 1997), DCRNN, STGCN, ASTGN, STSGCN, AGCRN. See Appendix A for more detail.

### 4.1 BASELINES

HA: Historical Average method. We use the average of the traffic over last hour to predict the next time slice.

ARIMA (Williams & Hoel, 2003): ARIMA is a well-known model for time series prediction.

LSTM (Hochreiter & Schmidhuber, 1997): LSTM can extract long-term and short-term temporal dependency and is widely used in time series forecasting.

DCRNN (Li et al., 2017): A deep learning framework for traffic forecasting that incorporates both spatial and temporal dependency in the traffic flow.

STGCN (Yu et al., 2017): STGCN uses spectral GCN and CNN to capture spatial and temporal dependencies respectively.

ASTGN (Guo et al., 2019): ASTGCN introduces a temporal and spatial attention mechanism to capture dynamic temporal and spatial correlations.

STSGCN (Song et al., 2020): STSGCN purely uses GCN to extract spatial-temporal information simultaneously.

AGCRN (Bai et al., 2020): AGCRN uses an adaptive graph convolutional recurrent network to capture node information.

## 4.2 COMPARISON AND RESULT ANALYSIS

Table 1 shows the overall prediction results including the average MAE, RMSE and MAPE of our proposed method and baselines. Due to the huge computation cost, it is hard to measure the performance of STSGCN on the Milan dataset. Compared with the models that can model spatial correlation, other models that only model temporal correlation (Historical Average method, ARIMA, LSTM) have poor performance in the three datasets. This is because such models ignore the spatial influence between nodes and only use the historical information of a single node. Among spatial-temporal models, ASTI-GCN achieves the best performance in all indicators except MAPE of PEMSD4 and MILAN dataset. Because STGCN, ASTGCN and DCRNN did not consider the heterogeneity in the spatial-temporal data, while STSGCN only considered the local spatial-temporal heterogeneity of three adjacent time steps, which was insufficient for the global information extraction, our model performed better. Besides, we find AGCRN performs well on road datasets, which is close to our results, but has poor performance on the mobile traffic dataset. We conjecture that the reason is large number of nodes and the large difference in node distribution in MILAN dataset. Meanwhile, it further indicates that ASTI-GCN has a stronger generalization ability than AGCRN.

Table 1: Performance comparison of different approaches(The best results are bold, the second best results are underlined).

| Dataset | PeMSD4 | | | PeMSD8 | | | Milan | | |
|---|---|---|---|---|---|---|---|---|---|
| Model | MAE | RMSE | MAPE | MAE | RMSE | MAPE | MAE | RMSE | MAPE |
| **HA** | 61.8 | 87.27 | 51.16 | 31.94 | 46.34 | 20.51 | 61.28 | 120.73 | 49.8 |
| **ARIMA** | 32.11 | 68.13 | 28.98 | 24.04 | 43.3 | 26.32 | 54.61 | 78.94 | 49.86 |
| **LSTM** | 29.04 | 45.63 | 40.71 | 25.93 | 38.64 | 20.77 | 42.93 | 80.29 | 56.33 |
| **STGCN** | 22.15 | 34.68 | 14.54 | 18.49 | 28.18 | 11.63 | 34.77 | 63.28 | 75.99 |
| **ASTGCN** | 22.99 | 35.42 | 15.75 | 18.89 | 28.64 | 12.64 | 39.42 | 72.04 | 38.17 |
| **DCRNN** | 22.23 | 35.25 | 14.78 | 16.45 | 25.41 | 10.73 | 33.9 | 65.52 | **17.01** |
| **STSGCN** | 21.29 | 33.81 | 13.87 | 17.18 | 26.85 | 10.91 | \ | \ | \ |
| **AGCRN** | 19.68 | 32.42 | **12.89** | 15.95 | 25.22 | 10.09 | 42.27 | 135.18 | 100.37 |
| **Ours** | **19.45** | **31.57** | 13.09 | **15.53** | **24.6** | **9.71** | **29.84** | **57.73** | 20.23 |

Figure 3 shows the changes of different metrics on the three datasets with the increase of predicted time steps. As we can see from the Figure 3, the prediction error increases over time, indicating that the prediction becomes more difficult. Those methods that only consider temporal correlation (ARIMA, HA, LSTM) perform well in short-term forecasting, but as the time interval increases their performance deteriorates sharply. The performance of GCN-based methods is relatively stable, which shows the effectiveness of capturing spatial information. Although our model has no outstanding performance in short-term forecasting tasks, it shows the best performance in medium and long-term forecasting tasks. This benefits from our spatial-temporal inception-GCN module which reduces the accumulation of errors significantly.

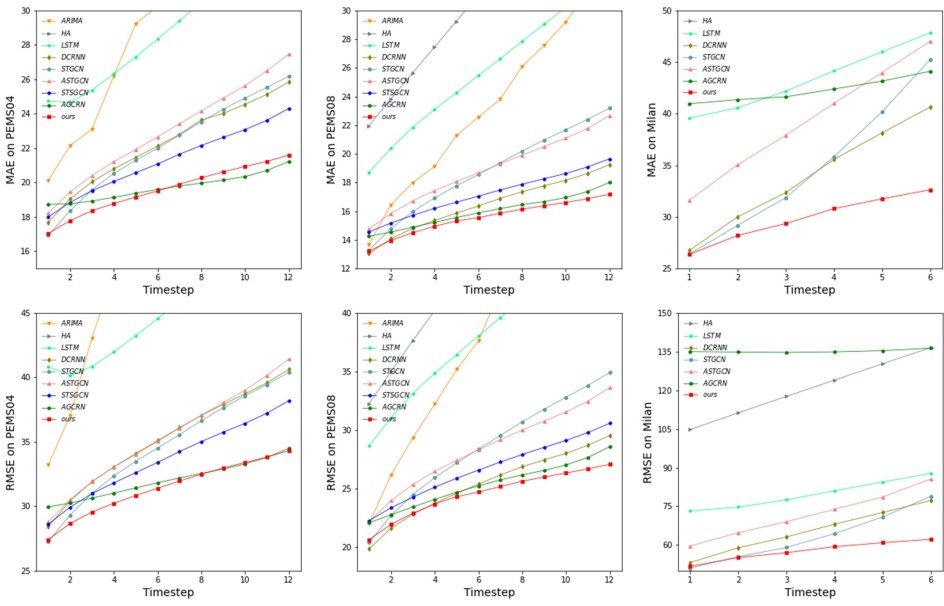

Figure 3: Prediction performance comparison at each time step.

## 4.3 COMPONENT ANALYSIS

In order to prove the effectiveness of each key component, we carried out component analysis experiments on PEMSD8 dataset. The basic information of each model is as follows:

Basic model: the model consists of two blocks. In each block, spatial and temporal convolution are performed respectively without the multi-scale spatial-temporal joint graph convolutions and graph attention. It does not use the sequence decoder with temporal attention as well. The output prediction result of the next time step simply uses a convolutional layer and the fully connected layers to generate.

+ STI-GCM: Based on the basic model, multi-scale spatial temporal joint graph convolutions are used to replace the separate spatial and temporal convolutions to extract spatial-temporal correlations. In addition, the graph attention is introduced to capture the heterogeneity.

+ Mask: We equip the basic model with the Mask matrix to learn the influence weights of different neighbors on the central node adaptively.

+ Sequence decoder with attention: We add to the basic model with the proposed sequence decoder and the temporal attention for multi-step forecasting.

+ alternant sequence decoder and short-term decoder: The output layer is modified into the fusion of the sequence forecasting result and short-term decoder forecasting result to benefit from both of them.

Except for the different control variables, the settings of each experiment were the same, and each experiment was repeated 10 times. The results which are shown in Table2 indicate that the model using STI-GCM has better performance, because the network can adaptively capture the heterogeneity of spatial-temporal network data. The sequence decoder with temporal attention shows good performance. The Mask mechanism also contributes some improvements. In the process of fusion, we train the short-term decoder and sequence decoder model circularly, and fuse the results of the two models as the final prediction output. It can not only ensure the short-term forecasting accuracy, but also avoid the influence of error accumulation on the long-term forecasting accuracy.

Table 2: Component Analysis

| Methods | MAPE | MAE | RMSE |
|---|---|---|---|
| basic | 11.25 | 18.69 | 28.32 |
| +STI-GCM | 10.64 | 17.36 | 26.96 |
| + Mask | 10.46 | 17.51 | 28.12 |
| + Sequence decoder with attention | 10.53 | 16.76 | 26.27 |
| + alternant sequence decoderand short-term decoder | 10.59 | 17.00 | 26.42 |
| ASTI-GCN | 9.71 | 15.53 | 24.60 |

## 5 CONCLUSION

This paper introduces a new deep learning framework for multi-step spatial-temporal forecasting. We propose spatial-temporal joint convolution to directly capture spatial-temporal correlation. At the same time, we employ the inception mechanisms to extract multi-scale spatial-temporal correlation, and introduce graph attention to model graph heterogeneity. Moreover, we combine short-term decoding and sequence decoding to map historical data to future time steps directly, avoiding the accumulation of errors. We evaluate our method on 3 real-world datasets from 2 different fields, and the results show that our method is superior to other baselines with good generalization ability.

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

## A APPENDIX

### A.1 DATASETS

We evaluate ASTI-GCN on two highway datasets (PEMSD4, PEMSD8) and one telecommunications dataset. PEMSD4 refers to the traffic data in San Francisco Bay Area from January to February 2018. PEMSD8 is the traffic data in San Bernardino from July to August 2016. The telecommunications dataset contains records of mobile traffic volume over 10-minute intervals in Milan. We summarize the statistics of the datasets in Table 3.

Table 3: Dataset description

| Datasets | Samples | Nodes | Timeslot | Input Length | Output Length |
|---|---|---|---|---|---|
| PEMSD4 | 16992 | 307 | 5min | 12 | 12 |
| PEMSD8 | 17856 | 170 | 5min | 12 | 12 |
| Mobile traffic | 4320 | 900 | 10min | 6 | 6 |

## A.2 ADJACENCY MATRIX DEFINITION

For the road traffic datasets, if two nodes are connected, the corresponding value in the adjacency matrix is set to 1, otherwise 0. The spatial adjacency matrix can be expressed as:

$$A_{i,j} = \begin{cases} 1, & \text{if } v_i \text{ connects to } v_j \\ 0, & \text{otherwise} \end{cases} \tag{9}$$

where $A_{i,j}$ is the edge weight of node $i$ and node $j$, $v_i$ is node $i$.

For the Milan dataset, Spearman correlation coefficient was used to define the adjacency matrix:

$$A_{i,j} = \begin{cases} 1, & i \neq j \text{ and Spearman} \left( t_{v_i}, t_{v_j} \right) > \delta \\ 0, & \text{otherwise} \end{cases} \tag{10}$$

where $t_{v_i}$ and $t_{v_j}$ are time series in training data of node **i** and node **j**, respectively. $Spearman(\cdot)$ is non-parametric indicator that measure the correlation between the two time series, $\delta$ is the threshold that controls the distribution and sparsity of the matrix, which is set to 0.92 in this article.

## A.3 METRICS

We evaluate the performance of our network with three evaluation metrics : Mean Absolute Error (MAE), Root Mean Square Error (RMSE) and Mean Absolute Percentage Error (MAPE).

$$MAE = \frac{1}{m} \sum_{i=1}^{m} |\hat{y}_i - y_i| \tag{11}$$

$$MAPE = \frac{100\%}{m} \sum_{i=1}^{m} \left| \frac{\hat{y}_i - y_i}{y_i} \right| \tag{12}$$

$$RMSE = \sqrt{\frac{1}{m} \sum_{i=1}^{m} (\hat{y}_i - y_i)^2} \tag{13}$$

where $\hat{y}_i$ and $y_i$ represent the predicted value and the ground truth respectively. $m$ is the total number of predicted values.

