# OpenReview forum: "Adaptive Spatial-Temporal Inception Graph Convolutional Networks for Multi-step Spatial-Temporal Network Data Forecasting"
_ICLR.cc/2021/Conference — Reject_

### Official Review · AnonReviewer2 · 2020-10-27
**The paper presents the Adaptive Spatial-Temporal Inception Graph Convolutional Networks for spatio-temporal forecasting. The motivation is clear and the contributions are incremental. However, the writing and experiments have a large space to improve.**

**Rating:** 3
**Confidence:** 5

**Review:**

This paper presents a GCN-based solution for multi-step spatio-temporal data forecasting. The main contributions are 1) a spatio-temporal joint graph convolution network to simultaneously capture spatial and temporal correlations; 2) a combination of the sequence decoder and short-term decoder to alleviate the error accumulation when modeling the temporal dependencies. I agree with the incremental contribution of this paper. To verify the effectiveness of their approach, the authors conduct experiments on three real-world datasets.

However, I have the following concerns:

Presentation:
The major concern is the writing of this paper. There are many parts (e.g., notations, statements) that are unreadable or hard to understand. For example,
1) What is “ST in GCM” in Figure 2(a)?
2) In page 4, x \in \mathbb{R}^n. What is n here? It seems to be inconsistent with the notation N in Section 3.1.
3) This paper introduces a learnable spatial mask matrix W_{mask}, but how to use it is not clear.
4) Does Figure2(c) in page 5 mean Figure 2(b)?
5) In the last equation of page 5 (for computing Y_{s_out}), what is t? Moreover, since W_F is of shape F by 1 and W_t is t by M, how can we multiply W_F by W_t?
6) How to obtain Y_{m_out} in equation 7?
7) Many typos and grammatical errors. I only point out several of them. Spatial-temporal data is -> are, correlations STG2Seq -> correlations. STG2Seq, Laplqaacian->Laplacian, which needed -> which is needed, we conjecture the reason is-> we conjecture that….

Technical:
1) Novelty of the proposed model is limited. The proposed spatio-temporal joint graph convolution is very similar to the concept of 3D GCN [1]. However, in this paper, I could not see any discussion about the difference and comparison between them.
2) In section 3.1, this paper assumes that the graph is undirected. However, the propagation of traffic in spatio-temporal domains are certainly directed (e.g., downstream, upstream).
3) What is the difference between the temporal attention in Section 3.4 and an FFN (two fully-connected (FC) layers)? In my view, this paper only uses two matrices for the dimension transformation, which is the same as two FC layers.

Experiments:
1) It is more reasonable to compare the proposed model with CNN-based solutions in the Mobile traffic dataset, as there is no explicit graph structure (only Euclidean structure exists). Baselines like ST-ResNet, ConvLSTM, DeepSTN+ or DeepLGR should be included for comparison.
2) According to the paper, I cannot see any detail of the multi-run experiments (e.g., in Table 1 and Figure 3). How about the stability (i.e., variance) of the proposed method?
3) The experiment shows little improvement to the existing approaches in the first two datasets. There should be significant test (e.g., student t-test) conducted to make sure the experimental result is reliable. Considering the limited improvement, it is also questionable whether adding such complexity to the model is worthy. It may not make sense to improve the MAE by 0.02 in the first dataset by increasing the runtime a lot.
4) No experiments to show the effects of the spatio-temporal kernels. Why should we set it 3x1, 1x3 and so on?
5) It seems that the MAPE in Table 1 largely exceeds 100%, which is unusual to see. What is the unit of MAPE here?

Minor issues:
1) All acronyms should be expanded for the first time in text (e.g., HA is not expanded. What is HA? Historical average?).
2) There should be one space before each citation. For example, DCRNN(Li et al., 2017) -> DCRNN (Li et al., 2017).
3) In the first paragraph of Section 1, the paper claims that “Typical applications include …, traffic road condition forecast (…, Liang et al., 2018), … and geo-sensory time series prediction (Li et al., 2017).” However, Liang et al., 2018 is for geo-sensory time series modeling while Li et al., 2017 is for traffic prediction. Please carefully read these papers before citing them.
4) It is difficult for those who are not familiar with the Inception Network to understand how your model draws insights from it. In addition, the paper proposing the Inception Network should be cited.
5) No future work discussed in this paper.

In summary, I recommend not to accept this paper in its present form.

Reference:
[1] Yu et al., 3D Graph Convolutional Networks with Temporal Graphs: A Spatial Information
Free Framework For Traffic Forecasting, Arxiv 2019.

---

> ### Author Response · Authors · 2020-11-24
> **Response**
>
> Dear AnonReviewer2, we thank you very much for your thorough review of our work and the positive comments. In the following we try to address all your concerns point by point.
>
> Presentation:
>
> 1. The blue block "ST in GCM" means "STI-GCM". We update the figure 2(a).
> 2. We correct the dimension of X.
> 3. To assign weights to different neighbors, we alse do the element-wise product between Wmask and adjacency matrix.
> 4. Thank you for pointing the mistake. We correct the Quoted image.
> We correct other typos in the paper.
>
> Technical:
>
> 1. Our method is different with the 3D GCN. We introduce node embedding and spatial-temporal inception-attention in our model.
> 2. Because most of datasets we used are undirected, we define the traffic network as undirected graph. In addition, spectral graph convolution is appropriate for undirected graph.
>
> Experiments:
>
> 1. Mobile traffic has noneuclidean structure, because each station like eNodeB has different neighbor stations.
> 4. We select the spatio-temporal kernels based on experiments.
> 5. MAPE is the mean((prediction - true)/true)*100%, so if the error is bigger than true value, the MAPE maybe bigger than 100%.
>
> We hope that we have addressed all your concerns and are grateful.
> References
> see paper

---

### Official Review · AnonReviewer4 · 2020-10-28
**Many method descriptions are unclear, and the writing needs to be substantially improved.**

**Rating:** 3
**Confidence:** 4

**Review:**

This paper proposes a spatial-temporal graph neural network, which is designed to adaptively capture the complex spatial-temporal dependency. Further, the authors design a spatial-temporal attention module, which aims to capture multi-scale correlations. For multi-step prediction instead of one-step prediction, they further propose the sequence transform block to solve the problem of error accumulations. The authors conducted experiments on three real-world datasets (traffic on highways and mobile traffic), which shows their method achieves the best performance.

Overall, I think the problem of capturing spatial-temporal dependency shown in Figure 1 is interesting. But the description of the method is unclear and the writing of this paper still needs improvements. Furthermore, I didn't find enough novelty that meets the standard of ICLR.

Concerns:

Methods:

I don't understand why the authors introduce spectral graph convolution networks while it seems that the authors only use GCN in the spatial domain.

The equation (4) is confusing: $X$ in the left changes to be $T_K(L)X$ without more explanations.

It's unclear about how to use the mask matrix $W_{mask}$ in the proposed model.

The equation $C_{gl} = \sum_{i=1}^T \sum_{j=1}^K C_{out}$ is confusing.

There are many method descriptions like the above examples, which the authors should describe more clearly.

Typos:

Page 3 Last Line: Laplqaacian

Page 5 Paragraph 2 Line 12: node $v_i$as

Page 8 Line 4: performed respectivelywithout

Page 8 Line 17: both of them..

---

> ### Author Response · Authors · 2020-11-24
> **Response**
>
> Dear AnonReviewer4, thank you very much for your valuable feedback. In the following, we try to address every raised concern and hope to meet your expectations.
> Methods:
> 1. We use graph Laplacians matrix to map x in spectral domain. But the computation of graph convolution is high, Chebyshev polynomial Tk(x) is used to approximate kernels.
> 2. Thank you for pointing the mistake. We correct the equation(4).
> 3. To assign weights to different neighbors, we alse do the element-wise product between Wmask and adjacency matrix.
> 4. Cout is the result of inception module, whose dimension is N*T*K*(F*B). Cgl is the result of global pooling with padding.
>
> In addition, we correct other typos in the paper.
>
> We hope that we have addressed all your concerns and are grateful.
>
> References
>
> see paper

---

### Official Review · AnonReviewer1 · 2020-10-29
**Official Blind Review #1**

**Rating:** 3
**Confidence:** 5

**Review:**

Summary:

The authors propose ASTI-GCN to solve the multi-step spatial-temporal data forecasting problem. The model uses a convolutional block to model the spatial-temporal correlations and an inception attention based module to capture the graph heterogeneity. They evaluate the proposed method on three different traffic prediction datasets.


Pros:
1. The problem of traffic prediction is important.

Cons:
1. The contributions are limited in this paper. The ideas of jointly model spatial-temporal information via convolutional layer (see 3D GNN to model irregular regions [1] and 3D CNN to model regular regions [2]), and multi-scale spatial-temporal modeling (see [3, 4]) are not new. These papers have been released for more than one year. Especially, the multi-scale motivations in [3] [4] are almost the same in the paper. The only difference is that this paper involves attention to weight different scales, which is also a very common practice. Thus, I think the contributions are not enough to be accepted by top machine learning conference.

2. The experiments could be improved.

- Besides the ablation studies in this paper (add modules to the base model), it would be more convincing to add the ablation studies by removing some components. Since only combining one module (results in Table 2) performs worse than AGCRN, removing some components and keep the rest could provide deeper analysis. In addition, it would be more convincing to conduct all the ablation studies on all three datasets.

- To support the claim of region heterogeneity, it would be more interesting to show some case studies to verify the motivation and see the reasons for the improvement. Otherwise, the improvement may come from the increasing of the number of parameters.

- It would be better to show the error bar for each result since the improvement in some datasets is limited (e.g., PeMS04).

3. Some figures could be improved. For example, some arrows in Figure 2(b) are broken.

[1] Yu, Bing, et al. "3d graph convolutional networks with temporal graphs: A spatial information free framework for traffic forecasting." arXiv preprint arXiv:1903.00919 (2019).

[2] Chen, Cen, et al. "Exploiting spatio-temporal correlations with multiple 3d convolutional neural networks for citywide vehicle flow prediction." 2018 IEEE international conference on data mining (ICDM). IEEE, 2018.

[3] Geng, Xu, et al. "Spatiotemporal multi-graph convolution network for ride-hailing demand forecasting." Proceedings of the AAAI Conference on Artificial Intelligence. Vol. 33. 2019.

[4] Cui, Zhiyong, et al. "Traffic graph convolutional recurrent neural network: A deep learning framework for network-scale traffic learning and forecasting." IEEE Transactions on Intelligent Transportation Systems (2019).

---

> ### Author Response · Authors · 2020-11-24
> **Official reply to Rev.1**
>
> We thank you for all the suggestions you proposed, while we think some comments are based on misunderstanding of our paper.
> 1. The ideas of our paper are like you mentioned [1][2][3][4]?
>
> Actually it’s not the same. Besides the [2][3][4] used separated parts to model the spatial and temporal correlations, the idea in [1] is a little near. But the 3D-convolution in [1] is not the actual spatial temporal joint convolution, since it can only model the sum of 0-(K-1) hops correlations in graph space. Our spatial temporal joint convolution approach is more flexible. We use the attention based multi-scale receptive filed aimed to capture the heterogeneity. The method is not complicated, but the effect is promising. Because the heterogeneity problem what we explained in introduction do exist, and we found one way to capture it. We found the nodes in the graph show different data intimacy which means they receive different intensities of spatial-temporal correlations (have different receptive filed) of the neighbors both in the spatial and temporal dimensions,. That’s one property of the spatial-temporal data show the heterogeneity what we tried to solve.
>
> 2. The suggestions with the experiments
>
> 	We would like to do deeper analysis and add the ablation studies on all three datasets. Since the time limits before, we would like to add them to show more credible experimental results data evidence.
>
> 	For region heterogeneity, we would like to think how to make it more convincing. We think your suggestions are really good. The paper has limited pages, we would like to adjust the structure of the article to display a lot of more effective information.
>
> 	We would like to know what kinds of error bar are needed, for example, the error bar with the different time steps forecast?
>
> Thank you so much for your guidance. We really appreciate your suggestions and time.

---

### Official Review · AnonReviewer3 · 2020-10-31
**Interesting idea of spatial-temporal joint graph convolution but need more justification**

**Rating:** 5
**Confidence:** 5

**Review:**

This paper investigates the important problem of spatial-temporal forecasting, and proposes a multi-scale spatial-temporal joint graph convolution that jointly model the heterogeneous spatial-temporal correlations. Empirical results on multiple real-world datasets shows promising results.

The draft seems to be written in a rush, with mistake/typos in paragraphs, figures and tables. Besides, the some of the major claims are not well supported by the empirical results. Here are details about the main concerns:

D1: Some major claims are not well supported by the experimental results.
- The main novelty of this paper the is  the spatial temporal joint convolution. Yet, its design needs more theoretical and empirically justifications.  The idea of the concatenating K order of Laplacian matrix is a bit ad-hoc lacking of theoretical justification.  Besides, the spatial temporal joint graph convolution is not necessary "more joint" than baseline algorithm like DCRNN which incorporates the graph convolution operation into the each sub-step of the RNN operation.
- Besides, the claimed convolution in the spectral domain is actually operations in the spatial domain since the final form of the convolution is essentially a polynomial of the Laplacian matrix without involved the transformation into the spectral domain.
- Limited ablation studies are conducted to show its effectiveness. There are some ablation study in Table 2, but not enough to well support the claims.  For example, what is performance gain by simplify replacing the graph convolution with the proposed spatial-temporal joint graph convolution? Is inception style convolution is actually needed? What is the different with and without it?
- Besides, as shown in Table 2, even after the adding the spatiotemporal convolution and the inception attention mechanism, the performance is still worse than some baselines, including DCRNN and AGCRN. Potentially it is also possible that the mask (which can be applied to other baselines) actually contributes to the major improvement.

D2: Presentation. This paper seems to be written in a rush, and the presentation need more polish.

- The technical part is a bit hard to follow. For example, when explaining the spatial-temporal joint convolution, it might be easier to understand if the author provides an additional figure about it, e.g., a 2-D matrix with size T x K, with filters of size K_t and K_s applied on it, where graph convolution is used for the feature extraction.
- Inconsistent name of the proposed model. The name of the proposed model is not consistent, for example it is called ASTI-GCN in abstract, while in Figure 2a it is called ATI-GCN. Yet, in Section 4 is is call ASTIGCN, and in Table 2 it is also called ours.
- Errors/typos in table and paragraphs. For example, in Table 1 the RMAE should RMSE. The Laplqaacian in Section 3.2 should be Laplacian. - In Equation (8), we may rewrite \hat{Y}_i as \hat{Y}(\Theta)_i so the loss is a function of the parameter \Theta.  In Section 4.2, we may Capitalize the first letter of alternative to make it consistent with the other three. Also remove the duplicate periods at the end of the sentence.
- In Equation (2), the T_k(L) probably represents the order k instead of order k-1 otherwise T_0(L) will become order -1.
- The datasets use, i.e., PeMS04, PeMS08, seems to be PeMSD4 and PeMSD8, we may keep it consistent with previous methods to facilitate comparison of metrics in baseline papers.
-  In the Table 2, it is mentioned that each setting has been run for 10 time, and it is helpful to also include the standard deviation to show the statistical significance of the improvement

---

> ### Author Response · Authors · 2020-11-24
> **Official reply to Rev.3**
>
> We truly appreciate your suggestions for our paper. With regard to your questions, below are some explanations:
> 1. Why spatial temporal joint convolution? Why not baseline algorithms like DCRNN?
> This paper was really written in a rush, so there are some ideas we did not express fully. Spatial temporal joint convolution is actually designed to capture the directly spatial-temporal joint correlations described in introduction. Baseline algorithms like DCRNN, use the separated part to build the spatial and temporal correlations separately, which cannot capture the directly spatial-temporal joint correlations in the scenes showed in Figure1 in our paper.
> 2. spatio or spectral domain?
> It’s spectral domain. We use the undirected adjacency matrix. And the Laplacian matrix has been rescaled by the largest eigenvalue of  .
> 3. Is inception style convolution is actually needed? What is the different with and without it?
> We use the inception for building the method to model heterogeneity. In more detail, we use the inception to capture different filed of receptive filed weights, combined with the graph node-level attention. Because we thought the heterogeneity behaves like different nodes are sensitive to different receptive fields. So this method actually can be called adaptive receptive fields spatial-temporal GCN.
> 4. Is the performance come from the mask?
> The performance is not come from the mask. We did plenty of experiments and the data we provided is real. The mask did not contribute a lot when we use it alone. We should design the ablation experiments carefully and to well support our innovation ideas.
>
> The presentation has been modified according to your suggestions. Thank you so much for your guidance

---

### Author Response · Authors · 2021-01-30
**Submission Withdrawn by the Authors**

I have read and agree with the venue's withdrawal policy on behalf of myself and my co-authors.

---

### Decision · Program_Chairs · 2021-01-07
**Final Decision**

**Decision:**

Reject

**Comment:**

The paper proposes a multi-scale spatial-temporal joint graph convolution for spatiotemporal forecastings. Many reviewers have concerns regarding novelty, baseline comparisons, and writing clarity of the draft.